# OpenReview forum: "HyperAgent: Generalist Software Engineering Agents to Solve Coding Tasks at Scale"
_ICLR.cc/2025/Conference — Submitted to ICLR 2025_

### Official Review · Reviewer_gPre · 2024-11-04

**Soundness:** 3
**Presentation:** 4
**Contribution:** 3
**Rating:** 6
**Confidence:** 4

**Summary:**

The paper focusses on the recently increasingly popular area of developing autonomous software agents for coding tasks. In particular, the paper focusses on the development of a software agent for generalizability across different tasks (github issue, repair etc) and languages (python, java). To this end, the paper proposes Hyper-Agent which uses four specialized agents - planner, navigator, code editor and finally code executor for scaling to generic SWE tasks. The planner agent acts as the main agent and is responsible to collaborating with multiple specialized agent in an asynchronous manner, in order to perform the final task. The specialized agents improve subtask execution performance while the asynchronous queue based execution improves throughput efficiency. Quantitative evaluations for three SWE benchmarks are provided in order to prove the efficacy of the proposed approach.

**Strengths:**

* The paper focusses on an important problem of developing generalist agents without being tailored for specific SWE or coding tasks (e.g., competition coding etc)
* The combination of subagents with asynchronous message queues for improving subtask efficiency and parallel execution (e.g., parallel editing across multiple files) is also interesting and novel
* Quantitative evaluations across multiple benchmarks are provided to demonstrate the generalizability across different coding tasks and programming languages.

**Weaknesses:**

While the key idea and problem is interesting, I have some questions regarding some experimental results and details,
* The idea of using the same base-LLM to act as different subagents has been previously explored before [1,2]
* How are the subagents specialized? Is the difference primarily form using different input contexts / system prompts, tools etc.
* The paper claims that the final agent is generalist and applicable across diverse tasks. However, its not clear if the context and implementation for each task is specialized or not?
      - For instance, when solving the swebench like tasks, the original swe-agent implementation provides task specific instructions / tips (e.g., reproducing the issue) in the input context. Is a similar strategy also applied for  hyper-agent?
* Also regarding the experimental evaluations, are the baselines and the proposed approach using the same base-llm (e.g, Tab. 1)?
    - In particular, according to Tab. 1, 7: the best performing hyperagent configuration using claude-3-sonnet, however some of the baseline results are reported with gpt4o.
* Also the numbers with some of the baselines seem off: autocoderover with gpt4o performs 30.67% on swebench-lite and 38.40% on swebench-verified (both higher than hyperagent). However, the reported numbers for the same in Tab. 1 seem to be much lower.
* Finally, while a minor point, I had some concerns regarding the usability in terms of avg. cost which is for instance 1.82 on swebench for the best hyperagent configuration. In contrast, a simple 3 stage model (localization, repair, rerank) with agentless achieves $0.34 per instance on average while achieving better solve-rate.
* How are the ablation studies performed in Sec. 6? For instance, w/o navigator which agent is used for code navigation the planner agent or any other subagent.
* Are there some incontext examples teaching which subagent should be called by the planner agent? If not, is it mostly reliant on the functon calling capability of the main planner agent?

References:

[1] Meta{GPT}: Meta Programming for A Multi-Agent Collaborative Framework, Hong et al, 2024

[2] OpenDevin: An Open Platform for AI Software Developers as Generalist Agents, Wang et al, 2024

**Questions:**

Please see weakness section above.

---

> ### Author Response · Authors · 2024-11-20
>
> Thank you for the thoughtful and detailed feedback on our manuscript. We appreciate your engagement with our work and the opportunity to clarify and strengthen our contributions. Below, we provide responses to the key points raised.
>
> ---
>
> ### Comment 1:
> > "The idea of using the same base-LLM to act as different subagents has been previously explored before [1,2].
> How are the subagents specialized? Is the difference primarily from using different input contexts / system prompts, tools, etc.?"
>
> We acknowledge the reviewer’s observation regarding prior work exploring the use of the same base LLM for different subagents. While this concept is not entirely novel, **HyperAgent** distinguishes itself through several key aspects:
>
> 1. **Dynamic Orchestration**:
>    Unlike prior work where subagents may operate in a rigid sequence, HyperAgent’s Planner dynamically decides which subagent to invoke based on the task requirements. This adaptability is key to handling complex workflows. Many additional details about how we enhance reasoning capabilities, save costs in the key component (Planner), and implement our other novelties are included in our response to Reviewer Wb3u.
>
> 2. **System Prompts and Contextual Specialization**:
>    Yes, each subagent in HyperAgent is specialized with role-specific prompts and tools. These prompts are carefully designed to align with the agent’s specialized tasks. For example:
>    - The **Navigator** focuses on exploring the codebase, retrieving relevant files, or locating dependencies.
>    - The **Editor** performs patch generation and iterative refinement, leveraging repair-specific tools.
>    - The **Executor** handles runtime validation and test execution.
>
>    This role differentiation allows HyperAgent to effectively decompose tasks into smaller, manageable units, enhancing overall system efficiency and performance.
>
> 3. **Tool Integration**:
>    Subagents interact with specialized tools that align with their roles, such as LSP-based tools for the Navigator and IDE-inspired editing features for the Editor. This modular integration ensures that each subagent operates with optimized resources for its respective function.
>
> ---
>
> ### Comment 2:
> > "The paper claims that the final agent is generalist and applicable across diverse tasks. However, it’s not clear if the context and implementation for each task is specialized or not?
> For instance, when solving the SWE-Bench-like tasks, the original SWE-Agent implementation provides task-specific instructions/tips (e.g., reproducing the issue) in the input context. Is a similar strategy also applied for HyperAgent?"
>
> **HyperAgent** is designed as a generalist system capable of addressing diverse software engineering tasks, while SWE-Agent’s approach could be considered more specialized. Here’s how they differ:
>
> #### **Dynamic Adaptation vs. Specialist Design**:
> - **SWE-Agent** relies on task-specific strategies tailored for GitHub issue resolution, such as pre-designed demonstrations that provide explicit instructions for reproducing issues or resolving them (as detailed in [SWE-Agent repository](https://github.com/princeton-nlp/SWE-agent/tree/main/trajectories/demonstrations)). This approach makes SWE-Agent highly effective for the GitHub issue task but also aligns it more closely with a specialist system, as it is designed specifically for a narrow set of tasks.
>
> - In contrast, **HyperAgent** dynamically adapts to tasks through its Planner, which interprets task prompts and constructs high-level strategies on the fly. For example, when solving SWE-Bench-like tasks, the Planner may decide to reproduce an issue based on dynamically extracted error traces, without relying on hardcoded, task-specific instructions. While integrating task-specific demonstrations like SWE-Agent could improve performance, HyperAgent’s focus remains on maintaining flexibility and generalizability across tasks.
>
> #### **Generalist Framework with Reusable Templates**:
> HyperAgent utilizes flexible task templates that abstract common software engineering challenges into reusable structures, such as patch generation or prediction tasks. These templates are populated with task-specific details (e.g., error traces for Defects4J or GitHub issues for SWE-Bench), enabling efficient adaptation to diverse tasks while minimizing task-specific design. This framework allows HyperAgent to handle new tasks with minimal reconfiguration, highlighting its generalist nature.
>
> ---
>
> By dynamically adapting to task requirements and leveraging reusable structures, HyperAgent demonstrates a **generalist approach**, in contrast to SWE-Agent’s specialist design, which is tightly coupled to the GitHub issue resolution task.

---

> > ### Author Response · Authors · 2024-11-20
> >
> > ### Comment 3:
> > > "Also regarding the experimental evaluations, are the baselines and the proposed approach using the same base-LLM (e.g., Tab. 1)?
> > In particular, according to Tab. 1, 7: the best-performing HyperAgent configuration uses Claude-3-Sonnet; however, some of the baseline results are reported with GPT4o."
> >
> > We appreciate the reviewer’s question regarding the consistency of the base-LLM used in evaluations. In **Tab. 1**, the configuration **HYPERAGENT-Full-2** is designed specifically to use the same base-LLM (GPT4o) as some of the baselines, ensuring a better **apple-to-apple comparison**. This approach provides a more equitable basis for evaluation.
> >
> > However, we observed **only minor performance differences** between HyperAgent configurations using **GPT4o** and **Claude-3-Sonnet**. The best-performing configuration (using Claude-3-Sonnet) reflects slightly improved performance in some tasks, but these differences do not significantly alter the overall conclusions of the study.
> >
> > ---
> >
> > ### Comment 4:
> > > "Finally, while a minor point, I had some concerns regarding the usability in terms of avg. cost, which is, for instance, 1.82 dollars on SWE-Bench for the best HyperAgent configuration. In contrast, a simple 3-stage model (localization, repair, rerank) without agents achieves 0.34 dollars per instance on average while achieving a better solve rate."
> >
> > The cost difference reflects **HyperAgent’s focus on scalability and adaptability**, enabling it to address **more complex and diverse software engineering tasks**. While the 3-stage agentless model achieves a lower cost per instance, it is optimized for **narrow, predefined workflows** and may not generalize effectively to **complex, real-world SE tasks** requiring dynamic reasoning and multi-agent collaboration (e.g., RepoExec, Defects4J). HyperAgent justifies its higher cost through its flexibility and broader applicability, as highlighted in our evaluation:
> >
> > 1. **Agentless vs. HyperAgent**:
> >    - **Agentless** models are cost-efficient but are not designed to handle the **dynamic reasoning and multi-agent collaboration** required for tasks like error reproduction, test generation, or complex patching workflows.
> >    - **HyperAgent**, by contrast, leverages a multi-agent architecture to delegate specialized tasks (e.g., navigation, repair, validation) to dedicated agents. This approach enhances adaptability across various scenarios, including tasks with **greater complexity**.
> >
> > 2. **Cost-Optimized Configurations**:
> >    HyperAgent’s **modular design** allows for the integration of smaller, cost-efficient models for specific roles like navigation. For instance:
> >    - In **SWE-Bench experiments**, we tested the **HyperAgent-Lite-1** configuration, which combines **Sonnet and Haiku 3**. This configuration achieved performance comparable to Agentless (25.33% vs. 24.30% on SWE-Bench Lite) while maintaining similar API costs.
> >    - This demonstrates HyperAgent’s ability to achieve **cost efficiency** without sacrificing significant performance.
> >
> > 3. **Future Cost Reduction Potential**:
> >    HyperAgent’s modularity also positions it for future improvements. With the emergence of **high-performance small models** like **Claude 3.5 Haiku** or **Qwen-2.5 7B**, we anticipate:
> >    - **Reduced costs** for HyperAgent configurations.
> >    - **Enhanced performance** by integrating these models into specific agent roles.
> >
> > ---
> >
> > By prioritizing **scalability, flexibility, and modularity**, HyperAgent is designed to address a wider range of tasks while maintaining the potential for future cost reductions as new, smaller LLMs are introduced. This approach ensures that HyperAgent is not only effective but also **adaptable to evolving computational and financial constraints**.

---

> ### Author Response · Authors · 2024-11-20
>
> ### Comment 5:
> > "How are the ablation studies performed in Sec. 6? For instance, w/o navigator which agent is used for code navigation: the planner agent or any other subagent?"
>
> In **Section 6**, we evaluated the system’s behavior when individual agents were removed, and their tasks were reassigned to other components. When an agent is removed, the **Planner** assumes the role of the missing agent by directly invoking tools to perform the tasks originally handled by the removed agent. For instance, without the **Navigator**, the Planner directly uses navigation tools to locate relevant files or dependencies. This increases context length and introduces noisy information, leading to performance degradation (as noted in our response to Reviewer Wb3u).
>
> ---
>
> ### Comment 6:
> > "Are there some in-context examples teaching which subagent should be called by the planner agent? If not, is it mostly reliant on the function-calling capability of the main planner agent?"
>
> We indeed provided some **in-context examples** for the Planner to demonstrate how to trigger and request specific subagents for their respective tasks. These examples significantly improved the Planner’s performance. Below is an example of one such in-context demonstration:
>
> ---
>
> #### **Example**:
> **Thought**:
> The issue is related to the `TimeSeries` object in the `astropy.timeseries` module. When a required column is removed from the `TimeSeries` object, the exception message is misleading. The expected behavior is to raise an exception that informs the user that required columns are missing. However, the actual exception message is confusing and does not clearly indicate the problem. To resolve this issue, I will first try to reproduce the bug by running the provided code snippet. Then, I will investigate the code in the `TimeSeries` class to see where the exception is being raised and why the message is misleading.
>
> **Agent Name**: Executor
> **Request**:
> Reproduce the bug by running the provided code snippet and verify that the exception message is indeed misleading. Please run the following code and report the output:
>
> ```python
> from astropy.time import Time
> from astropy.timeseries import TimeSeries
> import numpy as np
>
> time = Time(np.arange(100000, 100003), format='jd')
> ts = TimeSeries(time=time, data={"flux": [99.9, 99.8, 99.7]})
> ts._required_columns = ["time", "flux"]
> try:
>     ts.remove_column("flux")
> except ValueError as e:
>     print(e)
> ```
> **Context**:
>
> The TimeSeries class in the astropy.timeseries module is used for handling time-series data. It has a set of _required_columns that must always be present in the data. If a required column is removed, the system is expected to raise a clear exception informing the user about the missing columns. However, the current implementation raises a confusing or irrelevant exception message, which can mislead users and make debugging harder.

---

> > ### Author Response · Authors · 2024-11-25
> > **Reminder: Rebuttal Deadline Tomorrow**
> >
> > Dear Reviewer,
> >
> > We hope this message finds you well. This is a gentle reminder that the rebuttal deadline is tomorrow, November 26. We wanted to check if our response has sufficiently addressed your concerns. Please don’t hesitate to reach out if there are any remaining questions or points you’d like us to clarify or discuss further.
> >
> > Best regards,
> > The Authors

---

> > > ### Comment · Reviewer_gPre · 2024-11-27
> > >
> > > I thank the authors for their response, can the authors please also clarify regarding the following comment on the reported numbers in the paper:
> > > > Also the numbers with some of the baselines seem off: autocoderover with gpt4o performs 30.67% on swebench-lite and 38.40% on swebench-verified (both higher than hyperagent). However, the reported numbers for the same in Tab. 1 seem to be much lower.

---

> > > > ### Author Response · Authors · 2024-11-27
> > > >
> > > > Thanks for responding us, we use reported number from an independent reproduce attempt from OpenAI in following post [1] when they reproduced number of AutoCodeRover on SWE-Bench verified and lite version. Additionally, AutoCodeRover (v20240620) + GPT 4o (2024-05-13) results on SWE-Bench website are not fully verified (it did not have trajectories report). Therefore, we decided to stick to reproduced results for AutoCodeRover. Their official manuscript is also not updated with new experiments.
> > > >
> > > > [1] https://openai.com/index/introducing-swe-bench-verified/

---

> > > > > ### Author Response · Authors · 2024-12-02
> > > > > **Reminder: Rebuttal Deadline Tomorrow**
> > > > >
> > > > > Dear Reviewer,
> > > > >
> > > > > We hope this message finds you well. This is a gentle reminder that the rebuttal deadline is today, December 2. We wanted to check if our response has sufficiently addressed your concerns. Please don’t hesitate to reach out if there are any remaining questions or points you’d like us to clarify or discuss further.
> > > > >
> > > > > Best regards, The Authors

---

### Official Review · Reviewer_hBBP · 2024-11-04

**Soundness:** 2
**Presentation:** 3
**Contribution:** 2
**Rating:** 3
**Confidence:** 4

**Summary:**

This paper proposes a multi-agent framework to address general issues in software engineering development. The multi-agent system is designed with four components: an agent for planning implementation, an agent for retrieving file content within the project, an agent for making code modifications, and an agent for executing unit tests. The overall process control of these agents is managed by a planner, and corresponding APIs are defined for each agent to interact with the project. Experiments were conducted on three datasets: SWE-bench, RepoExec, and Defects4J. However, the final experimental results did not demonstrate that the proposed method is significantly stronger than other baseline methods.

**Strengths:**

1. This paper designs a multi-agent system with four components: an agent for planning implementation, an agent for retrieving file content within the project, an agent for making code modifications, and an agent for executing unit tests.
2. The overall process control of these agents is managed by a planner, and corresponding APIs are defined for each agent to interact with the project.
3. This paper conducts relatively solid experiments on three datasets.

**Weaknesses:**

1. Compared to previous work, such as ACR and SWE-Agent, the innovation in this paper is primarily the addition of an Execution Agent component; the other agents are almost identical to those in previous work.
2. The HypeAgent proposed in this paper does not show significant improvements over previous methods in the final experimental results.
3. The HypeAgent proposed in this paper does not compare its fault localization performance against SWE-Agent.

**Questions:**

1. For the executor, how does it find appropriate unit tests to execute? If defect fixing requires introducing unit tests that do not currently exist in the project, how does this agent function in such scenarios? This is a major shortcoming of the paper, and it lacks a detailed analysis in this area.
2. In Section 5.1, how does HypeAgent perform compared to other models on SWE-agent?
3. In Section 5.3, how do other open-source agents in Section 5.1 perform on this dataset?

---

> ### Author Response · Authors · 2024-11-22
>
> We appreciate the time and effort you have put into reviewing our manuscript and providing feedback. Below, we address your comments and concerns in detail:
>
> ---
>
> ### **1. Clarification on ACR**
>
> > "Compared to previous work, such as ACR and SWE-Agent, the innovation in this paper is primarily the addition of an Execution Agent component; the other agents are almost identical to those in previous work."
>
> Could you clarify what **“ACR”** refers to? If it is **AutoCodeRover (ACR)**, we would like to highlight that both **AutoCodeRover** and **SWE-Agent** are **not multi-agent frameworks**. Instead, they follow single-agent or pipeline designs. Here are some distinctions and novelties of **HyperAgent** compared to these systems:
>
> - **Freedom in Execution**:
>   - AutoCodeRover relies on rigid sequences of phases that are fixed for each task. If one phase fails (e.g., localization), the task cannot be completed.
>   - In contrast, HyperAgent’s **Planner** dynamically adapts to the task context, allowing it to revisit phases, correct mistakes, and explore alternative strategies.
>
> - **Faster Execution**:
>   - HyperAgent leverages **parallelism** and **asynchronous message queues**, significantly reducing execution times on large tasks compared to AutoCodeRover.
>
> - **Cost Efficiency**:
>   - SWE-Agent incurs prohibitively high costs due to long context windows and noisy interactions. HyperAgent achieves **similar or better performance** on benchmarks with significantly **lower API costs**.
>
> ---
>
> ### **2. Performance Improvements**
>
> > "The HyperAgent proposed in this paper does not show significant improvements over previous methods in the final experimental results."
>
> We respectfully disagree with this assessment and would like to provide clarification. **HyperAgent demonstrates substantial improvements over prior methods across multiple benchmarks, as detailed below**:
>
> - **RepoExec**:
>   - HyperAgent achieves **state-of-the-art results**, outperforming all existing baselines on repository-level code generation.
>
> - **Defects4J**:
>   - HyperAgent surpasses specialized baselines designed specifically for bug repair tasks, showcasing its **generalizability** and **scalability**.
>
> - **SWE-Bench**:
>   - While HyperAgent performs comparably on SWE-Bench Lite and Verified, its **broader applicability** and **modularity** make it a more versatile solution across tasks and programming languages.
>
> ---
>
> ### **3. Fault Localization Comparison with SWE-Agent**
>
> > "The HyperAgent proposed in this paper does not compare its fault localization performance against SWE-Agent."
>
> SWE-Agent does not natively support **Java**, making it incompatible with benchmarks like **Defects4J**. In contrast, HyperAgent’s modular design allows easy adaptation to new programming languages by swapping or extending its **language server backend**.
>
> **This adaptability is a key advantage of HyperAgent** over SWE-Agent, demonstrating its versatility in diverse programming environments. We will clarify this distinction in the manuscript to avoid any confusion.
>
> ---
>
> ### **4. Executor Design and Unit Tests**
>
> > "For the Executor, how does it find appropriate unit tests to execute? If defect fixing requires introducing unit tests that do not currently exist in the project, how does this agent function in such scenarios?"
>
> The **Executor** is designed to handle a wide range of tasks beyond simply executing unit tests:
>
> - **Unit Test Execution**:
>   - The Executor identifies existing test cases based on the Planner’s instructions, using **navigation tools** to locate test files in the codebase.
>
> - **Command Execution**:
>   - The Executor can execute complex commands, such as build scripts, installation processes, or testing pipelines.
>   - In case of failures, the Executor can debug the issue by iterating with the Planner.
>
> - **Test Creation**:
>   - When new unit tests are needed (e.g., for newly added features), the Executor can create and execute test cases based on the Planner’s instructions.
>
> We will expand the manuscript to provide more details about the Executor’s functionality and its ability to handle complex scenarios.
>
> ---

---

> ### Author Response · Authors · 2024-11-22
>
> ### **5. Performance Comparison on SWE-Agent**
>
> > "In Section 5.1, how does HyperAgent perform compared to other models on SWE-Agent?"
>
> Could you clarify this question? **SWE-Agent** is a system, not a benchmark.
> If you are referring to **SWE-Bench**, we have already compared HyperAgent to multiple baselines (e.g., HyperAgent-Full and HyperAgent-Lite configurations) in **Section 5.1**.
>
> ---
>
> ### **6. Comparison on Defects4J**
>
> > "In Section 5.3, how do other open-source agents in Section 5.1 perform on this dataset?"
>
> Could you clarify this question as well?
> We have compared all known agent systems on **Defects4J** at the time of writing this paper. If you are referring to SWE-Agent specifically, it does not support **Java**, making it incompatible with Defects4J.
>
> This limitation highlights **HyperAgent’s adaptability**, which supports both **Python (SWE-Bench)** and **Java (Defects4J)** out of the box.
>
> ---
>
> We hope these detailed responses address your concerns and clarify HyperAgent’s contributions and distinctions. We are committed to refining the manuscript further to incorporate these points and improve its clarity.
>
> ### **7. Additional Clarifications**
>
> Specialization of Subagents
>
> 	- Subagents are specialized using tailored prompts, tools, and reasoning capabilities. For example, the Navigator uses language servers for file retrieval, while the Editor integrates IDE-like editing tools to handle patch generation and refinement.
>
> ---
> We hope these responses address your concerns and clarify the contributions and novelties of our work. We will revise the manuscript to incorporate these points and ensure that HyperAgent’s design and results are clearly presented. Thank you again for your valuable feedback.

---

> > ### Author Response · Authors · 2024-11-25
> > **Reminder: Rebuttal Deadline Tomorrow**
> >
> > Dear Reviewer,
> >
> > We hope this message finds you well. This is a gentle reminder that the rebuttal deadline is tomorrow, November 26. We wanted to check if our response has sufficiently addressed your concerns. Please don’t hesitate to reach out if there are any remaining questions or points you’d like us to clarify or discuss further.
> >
> > Best regards,
> > The Authors

---

> > > ### Author Response · Authors · 2024-12-02
> > > **Reminder: Rebuttal Deadline Today**
> > >
> > > Dear Reviewer,
> > >
> > > We hope this message finds you well. This is a gentle reminder that the rebuttal deadline is today, December 2. We wanted to check if our response has sufficiently addressed your concerns. Please don’t hesitate to reach out if there are any remaining questions or points you’d like us to clarify or discuss further.
> > >
> > > Best regards, The Authors

---

### Official Review · Reviewer_B9Au · 2024-11-04

**Soundness:** 3
**Presentation:** 3
**Contribution:** 2
**Rating:** 6
**Confidence:** 3

**Summary:**

The paper introduces HyperAgent, a multi-agent system consisting of 4 primary agents (Planner, Navigator, Code Editor and Executor) organised in a Star-topology, to achieve generalizable software engineering capabilities across languages (Java and Python) and tasks (Issue-PR resolution over SWE-Bench, Bug localization and Repair over Defects4J, and repository-level code generation over RepoExec.). The planner agent creates subtasks and issues messages over async queues to each of the other 3 agents, which then try to execute the subtask. An LLM summarizer condenses the child agent trajectory to be sent back to the planner.

The different agents in the system can be instantiated with different LLMs, providing the ability to achieve a cost/capability tradeoff depending on the difficulty of the task.

The authors present results on 3 different software engineering tasks (end-to-end issue resolution, bug localization, bug repair, code generation) to demonstrate the generalizability of the proposed system.

**Strengths:**

The paper demonstrates a generalized software engineering agent. As such, generalizability of the approach, and strong demonstration (even if not state-of-the-art on individual benchmarks) across 3 different tasks is the main strength of the paper. The proposed multi-agent breakdown is simple, while achieving good performance, with ablations supporting the need for each component as well.

The paper also demonstrates the usefulness of multi-LLM multi-agent system through the HyperAgent-Lite configurations.

The application of dynamic message queues for asynchronous communication between the planner and child agents allows parallelizability and reduction in runtimes.

Further, the authors also identify the difficulty faced by LLM agents in using various IDE provided tools (like go_to_definition, which for example, actually reduced the resolve rate), and came up with augmentations to improve the tool (w/ search ablation in go_to_definition, etc.), which is instructive. This is particularly insightful, and points to further research in the direction of building better abstractions around existing code-tools that will be more suitable for LLMs.

The paper is written clearly, and easy-to-understand.

**Weaknesses:**

I have two primary concerns with the paper:
- *Effectiveness of multi-agent setup*: The proposed system drives the complete workflow with an LLM agent (the planner), which is in direct contrast to the proposal by Agentless, which achieves better results than the best HyperAgent configuration on SWEBench-Verified, while also being much more cost effective (\\$0.34 for Agentless vs \\$1.82 for HyperAgent). This calls into question whether a comparatively complex agentic system is required for the software engineering task.
- *Evaluation baseline for Defects4J* - I would suggest the authors compare HyperAgent to ChatRepair [1], which reports 114 solved cases on D4J v1.2. I seek to understand how HyperAgent compares to ChatRepair and if the proposed multi-agent approach offers benefits above the ChatRepair.

[1] Xia, C. S., & Zhang, L. (2023). Keep the Conversation Going: Fixing 162 out of 337 bugs for $0.42 each using ChatGPT. arXiv preprint arXiv:2304.00385.

**Questions:**

- Can the authors compare HyperAgent to Agentless (on SWEBench-Verified) and ChatRepair (on Defects4J)? Specifically, could you include average runtimes for Agentless and SWEAgent on SWEBench?
- For both RepoExec and Defects4J, can the authors include the results for at least one of the Full configurations? This will help with understanding the scalability of the approach to better models.

Minor clarifications which I believe could enhance the readability of the paper:

> For example, multiple Navigator instances can explore different parts of a large codebase in parallel, the Editor can apply changes across multiple files simultaneously, and the Executor can run tests concurrently, accelerating validation.

- Can the authors clarify if there are specific design decisions made to ensure synchronization between agents preventing them from modifying the same files?

> Failed tasks are requeued for reliability
> HYPERAGENT also has a problem of early exit (due to hallucination that the task has been
solved)

- As pointed out by the authors, it could be challenging for the LLMs to determine success/failure of task/subtasks. Can you clarify how are failures detected for requeuing?

- What are the details of HyperAgent-lite-3?

- Explored the sharing of previous context with a specific agent?

- Can the authors clarify if they intend to make the HyperAgent source code open source?

- Can the authors include a discussion about LSPToolKit and if there are specific design choices that were made to better align it as a tool for LLMs? Specifically, can the authors compare with some of the existing approaches that use LSP to build tools for LLMs. This could be instructive for devtool builders.

---

> ### Author Response · Authors · 2024-11-20
>
> Thank you for your thorough review and insightful feedback on our manuscript. We appreciate the opportunity to address your concerns and provide clarifications.
>
> ---
>
> ### Effectiveness of Multi-Agent Setup:
> We acknowledge the comparison with **Agentless**, which achieved notable results on SWE-Bench-Verified with a cost-effective approach. Our design of **HyperAgent** aims to balance **generalizability across diverse software engineering tasks** with **performance efficiency**. While Agentless demonstrates impressive results on specific benchmarks, its approach may not generalize as effectively to tasks requiring complex interactions, such as repository-level code generation or intricate bug localization.
>
> **HyperAgent’s multi-agent architecture** is designed to handle such complexities by delegating specialized tasks to dedicated agents, thereby enhancing adaptability across various scenarios. Additionally, we achieved **state-of-the-art (SoTA) results** on other benchmarks, even when compared to specialized and complex frameworks like **RepairAgent** on Defects4J.
>
> ---
>
> ### Comparison to ChatRepair:
> **ChatRepair** follows a fundamentally different evaluation procedure. It provides ChatGPT with “relevant information about the initial test failures as well as earlier patch attempts,” explicitly exposing the failure location to the model and requiring only **patch generation**.
>
> In contrast:
> - **HyperAgent** and other baselines (e.g., RepairAgent, SelfAPR) operate **fully autonomously**, starting from an error trace without prior localization information.
> - This makes the task significantly more challenging and emphasizes **end-to-end problem-solving capabilities**.
>
> While ChatRepair’s results are impressive, they do not reflect the **full pipeline of bug resolution**, making it unsuitable for direct comparison to HyperAgent. We acknowledge that this distinction could be clarified further in the manuscript, and we will update it to include these points.
>
> ---
>
> ### Comparison to Agentless on SWE-Bench-Verified:
> **HyperAgent** is already compared to Agentless on SWE-Bench-Verified, as shown in **Table 1**. While Agentless achieves a slightly higher task resolution rate (33.20% vs. 33.00% for HyperAgent-Full-1), **HyperAgent** provides a **modular architecture** suitable for **more complex tasks beyond SWE-Bench**, which Agentless is not designed to handle.

---

> > ### Author Response · Authors · 2024-11-20
> >
> > ### Comment 1:
> > > "Can the authors clarify if there are specific design decisions made to ensure synchronization between agents preventing them from modifying the same files?"
> >
> > **Synchronization** is managed by the **Planner**, which centralizes task generation and distribution. For example:
> > - The Planner ensures that tasks such as editing or navigating are assigned to distinct files or independent parts of the codebase.
> > - While intelligent planners like **Claude 3 Sonnet** rarely produce conflicting tasks, we include an additional **ad hoc check** to deduplicate tasks that overlap (e.g., editing the same file or issuing redundant navigation requests).
> > This design ensures **robust task coordination** and prevents errors.
> >
> > ---
> >
> > ### Comment 2:
> > > "As pointed out by the authors, it could be challenging for the LLMs to determine success/failure of task/subtasks. Can you clarify how failures are detected for requeuing?"
> >
> > **Failure detection** in child agents is relatively straightforward:
> > - Each tool explicitly reports errors (e.g., syntax issues or failed edits).
> > - These errors are summarized in the **agent trajectory** and sent back to the Planner.
> >
> > The Planner, however, faces challenges in determining whether the **entire task** is resolved. For example:
> > - In **SWE-Bench**, the Planner must analyze the output of reproduced scripts to decide whether they meet the task requirements.
> > - To address this, we allow the Planner to **repeat subtasks with additional instructions** when necessary.
> >
> > While this approach is not foolproof, it minimizes the likelihood of **premature task resolution**.
> >
> > ---
> >
> > ### Comment 3:
> > > "What are the details of HyperAgent-lite-3?"
> >
> > We apologize for the missing detail. The **HyperAgent-Lite-3** configuration is as follows:
> > - **Planner**: WizardLM2
> > - **Navigator**: Llama-3-8b
> > - **Editor**: WizardLM2
> > - **Executor**: Llama-3-8b
> >
> > ---
> >
> > ### Comment 4:
> > > "Can the authors clarify if they intend to make the HyperAgent source code open source?"
> >
> > We are committed to ensuring the **reproducibility** and **accessibility** of our research:
> > - The source code for an **early version of HyperAgent** has already been made publicly available in an open-source repository.
> > - However, in adherence to the **double-blind review process**, we cannot disclose the repository’s link at this time.
> >
> > Following the conclusion of the review process and **acceptance of the paper**, we will include the repository link in the **camera-ready version** for transparency and broader adoption.
> >
> > ---
> >
> > ### Comment 5:
> > > "Explored the sharing of previous context with a specific agent?"
> >
> > We include a **context field** in the requests sent by the Planner, which contains:
> > 1. **Objective Details**: Information necessary for the child agent to complete the specific task.
> > 2. **Global Background**: Broader context required for successful task execution (e.g., dependencies or impacted files).
> >
> > This approach helps save **cost** and **execution time** by enabling the Planner to:
> > - Notify child agents if there are instructions based on previous contexts.
> > - Transfer results from child agents to others when needed.
> >
> > For example:
> > - In earlier iterations, the **Navigator** may have explored the dependencies of a target function.
> > - In subsequent iterations, during the **editing phase**, the Planner will include some of these dependencies in its request to the **Editor**.

---

> > ### Comment · Reviewer_B9Au · 2024-12-02
> >
> > Thank you for addressing these concerns.

---

> ### Author Response · Authors · 2024-11-20
>
> ### Comment 6:
> > "Can the authors include a discussion about LSPToolKit and if there are specific design choices that were made to better align it as a tool for LLMs? Specifically, can the authors compare with some of the existing approaches that use LSP to build tools for LLMs? This could be instructive for devtool builders."
>
> We very appreciate the reviewer’s interest in **LSPToolKit**, as its development was a significant focus of our work earlier this year and even open-sourced, even preceding SWE-Agent. While many details about LSPToolKit are provided in **Appendix A.3**, we are happy to summarize some of the key design decisions that make it particularly effective as a tool for LLMs.
> To illustrate, let us take the example of the go_to_definition tool. In standard IDE functionality, go_to_definition requires the cursor position and the file containing the target word (e.g., a method call). Initially, we tasked the LLM with identifying the cursor position by providing the document content annotated with line numbers (similar to IDEs like VSCode). However, we observed that the LLM often struggled with accurately identifying the cursor position, particularly the line and column numbers.
> In subsequent iterations, we optimized this process by simplifying the input requirements. Instead of asking the LLM for both the cursor position and line number, we only requested the target word and the associated line number. This change significantly improved reliability, as the column number could be easily inferred through an exact match of the word within the specified line. Later, we further enhanced the tool by adding a lightweight line-search functionality based on the target word, which mitigates issues of LLM hallucination regarding line numbers.
> These iterative refinements showcase our emphasis on minimizing ambiguity and leveraging simple, deterministic mechanisms to reduce reliance on LLM reasoning for tool-specific functionalities. Compared to existing approaches that integrate Language Server Protocol (LSP) features with LLMs, LSPToolKit focuses on designing robust interfaces that reduce error-prone LLM decisions while maximizing alignment with standard IDE workflows. This philosophy ensures greater reliability and efficiency, particularly when handling complex real-world codebases.
> We agree that expanding on these comparisons and highlighting the lessons learned could be valuable for devtool builders. In future iterations, we will incorporate these insights into the main text to make the contributions of LSPToolKit clearer and more instructive.

---

> > ### Author Response · Authors · 2024-12-02
> > **Reminder: Rebuttal Deadline Today**
> >
> > Dear Reviewer,
> >
> > We hope this message finds you well. This is a gentle reminder that the rebuttal deadline is today, December 2. We wanted to check if our response has sufficiently addressed your concerns. Please don’t hesitate to reach out if there are any remaining questions or points you’d like us to clarify or discuss further.
> >
> > Best regards, The Authors

---

> > ### Comment · Reviewer_B9Au · 2024-12-02
> >
> > > We provided additional details in Appendix A.3 on how tools like LSPToolKit align with LLM capabilities and compared them with existing approaches.
> >
> > Thank you very much for including these details. The interventions introduced by the authors to make LSP responses more usable by LLM-based agents are well-motivated, and explained well. While studying the supplementary material submitted by the authors, I noticed that the authors build upon some past works in their source code, and I would encourage the authors to explicitly state the innovations built over-and-beyond the base frameworks used in this work. For example, the following works built frameworks to use LSPs to improve coding abilities of LLM-based systems:
> > * "Better context makes better code language models: A case study on function call argument completion." Proceedings of the AAAI Conference on Artificial Intelligence. Vol. 37. No. 4. 2023.
> > * "Monitor-guided decoding of code LMs with static analysis of repository context." Advances in Neural Information Processing Systems 36 (2024).
> > * "Copiloting the copilots: Fusing large language models with completion engines for automated program repair." Proceedings of the 31st ACM Joint European Software Engineering Conference and Symposium on the Foundations of Software Engineering. 2023.
> >
> > The last work among these specifically studies automated program repair, which is one of the three problems the author's study in this paper.

---

> > > ### Author Response · Authors · 2024-12-02
> > >
> > > Thank you for spotting out this, we will revise our references to include these works. We actually already included "Monitor-guided decoding of code LMs with static analysis of repository context." in our codebase readme in the open source version because LSPToolKit is based on multilspy for using multi languages LS functionalities. However, due to our mistakes, we did not cite this work explicitly, we will revise our manuscript to fix this error. Thank you again for your comments.

---

### Official Review · Reviewer_Wb3u · 2024-11-04

**Soundness:** 2
**Presentation:** 3
**Contribution:** 2
**Rating:** 5
**Confidence:** 4

**Summary:**

This work introduces HyperAgent, a multi-agent software engineering system that leverages Language Models. The HyperAgent system can be adapted to work on multiple Software Engineering benchmarks, specifically SWE-bench, RepoExec, and Defects4J. HyperAgent consists of one “parent” agent (the Planner) that can invoke one of three “child” agents (Navigator, Code Editor, Executor). These child agents can be invoked in parallel in any order by the Planner. The implementation also includes a suite of tools tailored to each child agent, along with flexible problem templating that allows for easy adaptation of HyperAgent to different coding tasks. HyperAgent performs comparably on several benchmarks, and the authors put forth several ablations examining how HyperAgent performs on such tasks.

**Strengths:**

The HyperAgent’s ability to generalize to multiple benchmarks is its main strong suite in my opinion. The adaptation of existing multi-agent methods to a larger span of software engineering tasks is great to see, and it is a good confirmation of the methodologies suggested by prior works. This work explicitly identifies clear stages of software engineering, then demonstrates how a multi-agent framework designed around these stages is extensible to many tasks. In addition to the implementation, HyperAgent demonstrates comparable performance to existing solutions across three benchmarks.

**Weaknesses:**

While the performance improvement of HyperAgent is admirable and the task coverage is quite impressive, I think it’s not quite clear what distinguishes HyperAGent from prior work. The methodology feels like a slight adaptation of existing multi-agent systems (e.g., MetaGPT) for the SWE-bench benchmark. The ablations, while interesting, look very similar to the ablations presented in SWE-agent. It is great that HyperAgent demonstrates comparable performance on several software engineering and coding benchmarks, and while this feels like a good engineering contribution, it remains unclear to me what new research questions or conclusions this exploration can offer up. I believe that more discussion about how HyperAgent’s architecture and methodology differs from prior work, along with analyses that back up these conclusions would make this paper a stronger research contribution.

- Section 2 Related Work: I would’ve liked more investment into a discussion of how HyperAgent differs from prior works, particularly in the second half of Section 2.2. After reading Sections 1 and 2, it is still not clear to me what new research questions HyperAgent addresses or what it does that truly distinguishes it from prior work.
- Section 3: Several comments - (1) How is this approach different from existing work such as MetaGPT, which arguably has more complex multi-agent systems following the Standard Operating Procedure framework? (2) Why is Multi-agent better than existing single agent approaches? How does having multiple agents lower inference costs or eliminate redundant information (wouldn’t the same information have to be communicated to multiple agents several times?) Also, it sounds like multiple child agents can be spawned in parallel - how is this more efficient than single agents, and why is the performance better because of this?
= The engineering details discussed in Section 3 sound quite impressive, particularly the use of the Message Queue and the general coordination of multiple agents. However, it is left to the reader to determine why these tools or multi-agent systems are interesting or novel relative to prior work. For instance, for Section 3.3, it is cool that these tools are made available, but unclear why this is better than the search and localization tools that prior works introduce (e.g. edit, search_dir, search_file in SWE-agent).
- Section 6 - These analyses are almost identical to the ones in SWE-agent. While this is interesting information, the paper may be stronger if there were more novel examinations that demonstrate how HyperAgent is distinct from prior work. For instance, “6.2 Analysis of Tool Design” is very similar to Table 3 in the SWE-agent paper. “6.3 Agent Behavior” is the same kind of graph as SWE-agent Figure 7. Analysis 6.1 is interesting, but it would be nice to know why the performance drops are not equal, and how certain tasks are still resolved even when there is no Editor, Navigator, or Executor.

**Questions:**

- Line 42: “their claim of addressing general software engineering tasks is overstated”. Why is this the case? Given that the downstream tasks you evaluate on are all evaluated on Python or Java, how does HyperAgent resolve the shortcomings of more versatility across tasks, languages, and dev. Scenarios?
- Lines 45 - 104: You introduce a workflow and claim that this is representative of most of software engineering - what evidence is there to reflect this? This seems very reminiscent of the issue resolution pipeline that prior works already introduce. Also, “Although different SE tasks require varied approaches, they all follow a similar workflow.” - Are there citations or prior evidence from existing SE works that reflect that the workflow you discuss is in fact the most reflective? This part of the introduction feels quite anecdotal.
- Line 99 - How are your three advantages made clear in your evaluation? Aside from higher performance, I do not see why existing works are less generalizable than HyperAgent, or why these agents are more efficient or scalable.
- Line 116 - What is SWE-bench-Python, RepoExec-Python, and Defects4J-Java? Are these variants of the original benchmark?
- Line 118 - SWE-agent and OpenDevin were both evaluated on non SWE-bench tasks, including HumanEval, HumanEvalFix, Bird, BioCoder, WebArena, and much more. The claim that this is the first system to work on many SE tasks feels like an over-claim.
- Line 254 - “A task can *be* defined…”
- Line 251 - Missing space between “HyperAgent” and “is”
- Line 328 - What does processing time account for? Does this include or exclude the calls to the API? What’s the variance on these numbers?

---

> ### Author Response · Authors · 2024-11-20
>
> We sincerely appreciate your thoughtful and detailed feedback on our manuscript. Your insights have been invaluable in helping us clarify our contributions and strengthen the presentation of our work. Below, we address your concerns point by point.
>
> ---
>
> ### Comment:
> > "It’s not quite clear what distinguishes HyperAgent from prior work. The methodology feels like a slight adaptation of existing multi-agent systems (e.g., MetaGPT) for the SWE-bench benchmark."
>
> **HyperAgent significantly diverges from existing multi-agent systems through its design philosophy and capabilities, addressing tasks that require dynamic reasoning, scalability, and specialized workflows.**
>
> #### **1. Scope and Workflow Design**:
> Current multi-agent systems like MetaGPT Primarily focuses on constructing entire software projects from scratch (e.g., building a simple program or repository). Its rigid waterfall-like workflow is optimized for end-to-end software construction, following predefined Standard Operating Procedures (SOPs) without dynamic adaptability. In contrast, HyperAgent  focuses on solving a broader range of challenging and real-world software engineering (SE) tasks. These include bug localization, fault repair, and repository-level code generation—tasks that require dynamic reasoning, fine-grained localization, iterative editing, and scalable execution. This requires a planner-centered framework, not just a procedural SOP-following system. **Unlike MetaGPT, HyperAgent’s Planner is dynamically adaptive, orchestrating agents based on task requirements, parallelizing subtasks when possible, and revisiting earlier phases (e.g., re-localization if editing fails)**. This makes HyperAgent more suitable for tasks requiring iterative problem-solving, rather than rigid task decomposition.
>
> ---
>
> #### **2. Agent Design for Specialized SE Tasks**:
> HyperAgent introduces specialized tools and **base LLM assignments** tailored for complex SE tasks.
> - Unlike **MetaGPT**, which relies on **prompt customization**, HyperAgent assigns specialized roles (e.g., navigation, editing, execution) and optimizes tools and suggests suitable LLMs for these roles, ensuring scalability, precision and cost saving in solving SE challenges.
>
> ---
>
> #### **3. Scalability Through Parallel Execution**:
> HyperAgent implements **asynchronous communication** and **parallel task execution** via a message queue system, a capability absent in MetaGPT and SWE-Agent.
> - For example, in tasks like **deploying a web app**, HyperAgent can handle subtasks such as **deployment scripts**, **CI/CD configurations**, and **documentation preparation** simultaneously, reducing overall execution time.
> - In SWE-Bench, this scalability is evident: HyperAgent achieves time efficiency (106–320 seconds vs. AutoCodeRover’s 720 seconds).
>
> ---
>
> #### **4. Planner-Centered Reasoning for Complex SE Tasks**:
> HyperAgent’s Planner surpasses MetaGPT’s static Manager by **actively monitoring task progress**, adjusting strategies, and revisiting prior phases as needed.
> Key design features include:
> - **Minimized Contextual Noise**: Child agents transfer only essential information to the Planner, avoiding noisy contexts that degrade LLM reasoning capabilities, as noted in [1].
> - **Format-Free Reasoning**: The Planner operates without rigid agentic formats, enabling better reasoning and problem-solving, as supported by [2].
>
> ---
>
> **In summary, HyperAgent is not a slight adaptation of MetaGPT but a dynamically adaptive framework built for real-world, complex SE tasks, incorporating specialized agents, parallel execution, and iterative problem-solving strategies.**
>
> [1] Fraga, Natanael. "Challenging LLMs Beyond Information Retrieval: Reasoning Degradation with Long Context Windows." (2024).
>
> [2] Tam, Zhi Rui, et al. "Let me speak freely? a study on the impact of format restrictions on performance of large language models." arXiv preprint arXiv:2408.02442 (2024).

---

> > ### Author Response · Authors · 2024-11-20
> >
> > ### Comment:
> > > "Section 2 Related Work: I would’ve liked more investment into a discussion of how HyperAgent differs from prior works, particularly in the second half of Section 2.2. After reading Sections 1 and 2, it is still not clear to me what new research questions HyperAgent addresses or what it does that truly distinguishes it from prior work."
> >
> > **HyperAgent addresses the challenge of designing a scalable, cost-effective, and generalizable multi-agent system for tackling complex software engineering (SE) tasks across diverse domains.**
> >
> > ---
> >
> > ### Key Research Questions Addressed:
> > 1. **Balancing advanced reasoning with cost-efficient execution**:
> >    How can we design a multi-agent system that balances the need for advanced reasoning with cost-efficient execution?
> > 2. **Seamless transfer across diverse SE tasks**:
> >    How can this system seamlessly transfer across different SE tasks, including fault localization, repository-level code generation, and bug repair?
> >
> >
> > As of the writing of this paper, high-performance agent systems on SWE-Bench either suffer from highly restricted and task-specific designs that limit their adaptability to other tasks (e.g., AutoCodeRover, Agentless) or incur prohibitively high operational costs (e.g., SWE-agent). Therefore, we believe that a system that has the centralization of advanced reasoning in the Planner agent with delegation of computationally intensive but conceptually simpler tasks to specialized child agents equipped with specialized tools is our key solution (1st passage in Section 3)
> >
> > ---
> >
> > ### HyperAgent’s Contributions to Multi-Agent Frameworks:
> > #### **1. Centralized Reasoning in the Planner**:
> > HyperAgent introduces the **Planner** as a central reasoning engine:
> > - **Dynamic strategy adjustment**: The Planner dynamically adjusts strategies and coordinates subtasks.
> > - **Efficiency and precision**: Unlike **MetaGPT’s Manager**, which statically delegates SOPs, or **SWE-Agent**, which makes repeated expensive LLM calls while handling long and noisy tool observations, the Planner consolidates reasoning to minimize unnecessary overhead.
> >
> > ---
> >
> > #### **2. Scalability and Efficiency**:
> > HyperAgent integrates a **message queue system** to enable parallel execution of subtasks by child agents:
> > - **Parallelism for scalability**: This design allows large tasks to be divided and executed concurrently, reducing overall execution time.
> > - **Improved efficiency**: In contrast, **SWE-Agent’s sequential architecture** and **MetaGPT’s waterfall-like workflow** limit scalability and extend execution times.
> >
> > ---
> >
> > #### **3. Generalizability via Modular Design**:
> > HyperAgent employs a **modular design** that allows seamless adaptation to new tasks, programming languages, or domains:
> > - **Specialized tools and task template prompts**: HyperAgent uses tools for navigation, editing, and execution, abstracted through task templates (Section 3.4).
> > - **Natural interfaces for language servers**: Tools like linting, “go to definition,” and code search are integrated via language servers, which can be easily swapped or expanded to support new programming languages.
> > - **Effortless extensibility**: Unlike SWE-Agent, which relies on bash commands as tools, HyperAgent’s use of open-source language servers simplifies the addition of new languages and features.
> >
> > **These modularity and abstraction mechanisms make HyperAgent uniquely adaptable to new programming languages, tasks, or domains.**
> >
> > ---
> >
> > **In summary, HyperAgent distinguishes itself by addressing limitations in prior frameworks through centralized reasoning, scalability via parallelism, and a modular architecture that ensures cost efficiency and broad applicability.**

---

> > > ### Author Response · Authors · 2024-11-20
> > >
> > > ### Comment:
> > > > "(2) Why is Multi-agent better than existing single agent approaches? How does having multiple agents lower inference costs or eliminate redundant information (wouldn’t the same information have to be communicated to multiple agents several times?)"
> > >
> > > The **multi-agent system** in HyperAgent is designed to balance **reasoning efficiency** with **cost-effective execution**. Here’s why this approach is advantageous compared to existing single-agent methods:
> > >
> > > ---
> > >
> > > #### **1. Delegation of Tasks**:
> > > - The multi-agent design allows the **Planner agent** to focus exclusively on **high-level reasoning and coordination**, while delegating computationally intensive but simpler tasks (e.g., code search or test execution) to lightweight child agents.
> > > - For example:
> > >   - The **Navigator agent** efficiently retrieves contextual information without requiring the Planner to engage in low-level operations.
> > > - This **division of labor** optimizes the system’s overall efficiency.
> > >
> > > ---
> > >
> > > #### **2. Streamlined Communication**:
> > > - Child agents share **only essential results** (e.g., localized bug locations or edited code snippets) with the Planner, rather than raw or verbose outputs.
> > > - This reduces computational overhead and allows the Planner to work with **concise, information-dense inputs**, which improves the reasoning performance of LLMs.
> > > - As demonstrated in **Appendix A.8**, the Navigator exchanges only the necessary information with the Planner, instead of exposing it to the lengthy and detailed outputs from tools like SWE-Agent.
> > >
> > > ---
> > >
> > > #### **3. Parallelism for Scalability**:
> > > - **Multi-agent parallelism** enables efficient handling of large-scale tasks.
> > > - For example:
> > >   - Multiple **Navigator instances** can explore different sections of a large codebase simultaneously.
> > >   - Multiple **Editor agents** can independently resolve subtasks, such as fixing separate bugs.
> > > - This **parallel processing** significantly speeds up execution without duplicating work or wasting resources.
> > >
> > > ---
> > >
> > > **In summary**, the multi-agent approach in HyperAgent ensures better task delegation, streamlined communication, and scalable parallel processing, making it more efficient and cost-effective than single-agent methods.

---

> > > > ### Author Response · Authors · 2024-11-20
> > > >
> > > > ### Comment:
> > > > > "But unclear why this is better than the search and localization tools that prior works introduce (e.g., edit, search_dir, search_file in SWE-agent)."
> > > >
> > > > **HyperAgent improves upon prior search and localization tools by employing a more modular and extensible approach, enabling advanced and efficient navigation.**
> > > >
> > > > ---
> > > >
> > > > #### **1. Modular and Language-Agnostic Framework**:
> > > > - Unlike rigid tool sets like **edit**, **search_dir**, and **search_file** in SWE-agent, HyperAgent leverages **language server-based interactions**.
> > > > - This abstraction provides a **dynamic and language-agnostic framework**, allowing for seamless adaptability to different programming languages and environments.
> > > >
> > > > ---
> > > >
> > > > #### **2. Advanced Functionalities**:
> > > > HyperAgent supports enhanced capabilities, including:
> > > > - **Proximity-Based Keyword Searches**: Improves accuracy by focusing on contextually relevant parts of the codebase.
> > > > - **Ranked Symbol Prioritization**: Orders results by relevance to streamline localization tasks.
> > > > - **Contextual Previewing**: Displays surrounding context to reduce unnecessary tool invocations.
> > > >
> > > > ---
> > > >
> > > > #### **3. Limitations and Future Improvements**:
> > > > - While Section 6.2 includes an **ablation analysis of tool design**, we acknowledge that the manuscript does not yet include a direct comparison with SWE-agent’s tools.
> > > > - We will enhance the manuscript to include a detailed comparison with SWE-agent’s tools to better showcase the effectiveness of HyperAgent’s approach.

---

> > > > > ### Author Response · Authors · 2024-11-20
> > > > >
> > > > > ### Comment:
> > > > > > "Section 6 - These analyses are almost identical to the ones in SWE-agent. While this is interesting information, the paper may be stronger if there were more novel examinations that demonstrate how HyperAgent is distinct from prior work. For instance, '6.2 Analysis of Tool Design' is very similar to Table 3 in the SWE-agent paper. '6.3 Agent Behavior' is the same kind of graph as SWE-agent Figure 7. Analysis 6.1 is interesting, but it would be nice to know why the performance drops are not equal, and how certain tasks are still resolved even when there is no Editor, Navigator, or Executor."
> > > > >
> > > > > We appreciate the reviewer’s observation and agree that further analysis could strengthen the paper. **While Section 6.1 aimed to explore how the Planner interacts with the system when individual agents are removed, deeper insights into the Planner’s behavior and its enhanced reasoning without specialized agents/tools would better highlight HyperAgent’s advantages.**
> > > > >
> > > > > ---
> > > > >
> > > > > #### **Details of the Section 6.1 Analysis**:
> > > > > - **Setup**: When an agent is removed, the **Planner** assumes its responsibilities by directly invoking the agent’s tools.
> > > > >   - This increases the Planner’s **context length**, introducing irrelevant and noisy information that can hinder its reasoning capabilities.
> > > > >
> > > > > ---
> > > > >
> > > > > #### **Why Performance Drops Vary**:
> > > > > 1. **Navigation Tasks**:
> > > > >    - Navigation tasks tend to generate highly exploratory outputs, such as large file lists or unrelated code snippets.
> > > > >    - These outputs flood the Planner’s context with unnecessary details, significantly degrading performance.
> > > > >
> > > > > 2. **Editing Tasks**:
> > > > >    - Editing tasks often involve **repeated tool calls** during patch refinement.
> > > > >    - This introduces **redundant information** into the Planner’s input, leading to cognitive overload.
> > > > >
> > > > > 3. **Task Complexity Differences**:
> > > > >    - These differences in task complexity and output characteristics explain why performance drops vary when specific agents are removed.
> > > > >    - For example, navigation tasks affect performance more due to their **exploratory and noisy nature**, while editing tasks add redundancy.
> > > > >
> > > > > ---
> > > > >
> > > > > #### **How Certain Tasks Are Still Resolved**:
> > > > > - Even without specialized agents, the **Planner’s reasoning capabilities** and access to tools allow it to resolve some tasks.
> > > > > - However, this comes at the cost of reduced efficiency and increased noise, which highlights the critical role of specialized agents in maintaining focused and clean contexts.
> > > > >
> > > > > ---
> > > > >
> > > > > **In summary**, by delegating exploratory or repetitive tasks to specialized agents, HyperAgent ensures a cleaner and more focused context for the Planner. This leads to **better reasoning, higher task success rates**, and a significant advantage over prior systems. We will refine our manuscript to further emphasize these findings and provide additional analyses to clearly demonstrate HyperAgent’s unique contributions.

---

> ### Author Response · Authors · 2024-11-20
>
> ### Comment:
> > "Line 42: 'Their claim of addressing general software engineering tasks is overstated.' Why is this the case? Given that the downstream tasks you evaluate on are all evaluated on Python or Java, how does HyperAgent resolve the shortcomings of more versatility across tasks, languages, and dev. scenarios?"
>
> **We acknowledge that our evaluation focuses on Python (SWE-Bench, RepoExec) and Java (Defects4J), but HyperAgent is designed as a flexible and extensible framework capable of supporting additional languages and scenarios in the future.**
>
> #### **Limitations of Current Evaluation**:
> - Demonstrating capability across all programming languages is **beyond the scope of this work**, as it would require more engineering efforts to evaluate on other repository-level tasks (these kinds of benchmarks are still rare today).
> - Current benchmarks, such as Python-based SWE-Bench and RepoExec, are widely adopted in the research community due to Python’s popularity.
>
> #### **Demonstration of Generalization**:
> - We have demonstrated HyperAgent’s **adaptability** by extending its capabilities from **Python** (SWE-Bench, RepoExec) to **Java** (Defects4J).
> - This showcases its potential for generalization across **different programming languages**, even within the scope of our current evaluation.
> - HyperAgent’s modular design, leveraging **language server-based tools**, makes it straightforward to expand support to additional languages.
>
> ---
>
> ### Comment:
> > "Line 99 - How are your three advantages made clear in your evaluation? Aside from higher performance, I do not see why existing works are less generalizable than HyperAgent, or why these agents are more efficient or scalable."
>
> **HyperAgent’s flexibility, efficiency, and scalability are reflected in its design and demonstrated in its evaluation, even if not fully elaborated.**
>
> #### **1. Generalizability**:
> - **Flexibility Across Tasks**: HyperAgent seamlessly handles **fault localization**, **repository-level code generation**, and **bug repair**, unlike task-specific systems such as AutoCodeRover and Agentless.
> - **Cross-Language Support**: Extending from Python to Java in our evaluation (Defects4J) demonstrates HyperAgent’s potential for generalization across languages.
>
> #### **2. Efficiency**:
> - **Streamlined Communication**: HyperAgent uses concise interactions between agents and the Planner, reducing the computational overhead seen in prior frameworks like SWE-Agent.
> - **Example**: In SWE-Bench, HyperAgent efficiently solves tasks with **lower computational overhead**, aided by its modular and lightweight agent design.
>
> #### **3. Scalability**:
> - **Parallelism**: HyperAgent employs **multi-agent parallelism**, where subtasks (e.g., navigation, editing) can be executed concurrently.
> - **Example**: In large-scale tasks, HyperAgent significantly reduces execution time by delegating work across multiple agents, as seen in the faster resolution of SWE-Bench tasks compared to sequential architectures like SWE-Agent.

---

> ### Author Response · Authors · 2024-11-20
>
> ### Comment:
> > “Line 116 - What is SWE-bench-Python, RepoExec-Python, and Defects4J-Java? Are these variants of the original benchmark?”
>
> No, these are not variants of the original benchmarks. The format **BenchmarkName-Language** is our way of specifying the programming language associated with each benchmark, but we understand that this may have caused some confusion. For example, SWE-Bench focuses on resolving GitHub issues for Python projects, so we refer to it as SWE-Bench-Python [3]. Similarly, RepoExec [4] is a repository-level code completion benchmark for Python repositories, so we denote it as RepoExec-Python. Defects4J is a widely used benchmark for bug detection and program repair for Java programs, so we refer it to Defects4J-Java [5].
>
> We sincerely apologize for any confusion caused by this naming convention and will make sure to clarify this more explicitly in the paper. This notation was intended to highlight the range of software engineering tasks HyperAgent can handle, demonstrating its capability to work at a repository-level scale across different programming languages.
>
>
> [3] Jimenez, Carlos E., et al. "SWE-bench: Can Language Models Resolve Real-world Github Issues?." The Twelfth International Conference on Learning Representations.
>
> [4] Hai, Nam Le, Dung Manh Nguyen, and Nghi DQ Bui. "REPOEXEC: Evaluate Code Generation with a Repository-Level Executable Benchmark." arXiv preprint arXiv:2406.11927 (2024).
>
> [5] Just, René, Darioush Jalali, and Michael D. Ernst. "Defects4J: A database of existing faults to enable controlled testing studies for Java programs." Proceedings of the 2014 international symposium on software testing and analysis. 2014.

---

> ### Author Response · Authors · 2024-11-20
>
> ### Comment:
> > “Line 118 - SWE-agent and OpenDevin were both evaluated on non SWE-bench tasks, including HumanEval, HumanEvalFix, Bird, BioCoder, WebArena, and much more. The claim that this is the first system to work on many SE tasks feels like an over-claim.”
>
> Thank you for your comment. ***Our work specifically targets repository-level software engineering tasks, which we believe are essential for addressing real-world programming challenges. These tasks involve fixing bugs or completing functions in actual software repositories and demand navigating and modifying interconnected codebases, resolving cross-file dependencies, and running comprehensive tests***. Such tasks are significantly more complex and realistic than those defined in the benchmarks you mentioned, which are primarily designed to evaluate isolated capabilities of LLMs.
>
> For instance, benchmarks like HumanEval and HumanEvalFix are valuable for assessing the ability of models to generate code snippets but do not represent the intricacies of software engineering. Real-world software engineering rarely involves single-file or self-contained problems; instead, it requires iterative problem-solving across multiple files, maintaining coherence throughout the codebase, and adapting to the context of large, interconnected systems.
>
> Repository-level benchmarks, such as RepoExec and Defects4J, better reflect these real-world scenarios. They simulate workflows that developers encounter daily, requiring deep interactions with the structure and context of repositories. In contrast, benchmarks like HumanEval, Bird, BioCoder, and WebArena focus on simpler, narrowly scoped tasks, such as generating code from concise prompts. While these benchmarks are useful for evaluating fundamental capabilities, they fail to capture the full spectrum of challenges inherent to real-world software engineering.
>
> Our work bridges this gap by demonstrating strong performance on repository-level tasks, showcasing HyperAgent’s ability to handle complex, real-world scenarios. These benchmarks push systems beyond generating isolated code, instead focusing on solving end-to-end tasks at scale. This distinction highlights the novelty of our approach and sets it apart from systems evaluated on benchmarks that do not reflect the complexities of real-world software engineering.
>
> We will revise our manuscript to clarify this distinction and contextualize our contributions more effectively, emphasizing the unique challenges addressed by HyperAgent and its capabilities in tackling interaction-heavy, repository-level software engineering tasks.

---

> > ### Author Response · Authors · 2024-11-20
> >
> > ### Comment:
> > > “Line 328 - What does processing time account for? Does this include or exclude the calls to the API? What’s the variance on these numbers?”
> >
> > The processing time we report accounts for all activities required to resolve a task. This includes tool initiation, tool invocation, and calls to APIs for external computations. Specifically, it measures the end-to-end time taken from task initialization to resolution, capturing the entire workflow of HyperAgent.
> >
> > For SWE-Bench Lite and Verified and Lite-1 configuration, the average processing time was 106 seconds, with a standard deviation of 21 seconds.
> >
> > We can include these details in the manuscript to clarify how processing time is measured and the variability observed in different scenarios.

---

> > > ### Author Response · Authors · 2024-11-20
> > >
> > > ### Comment:
> > > > “Lines 45 - 104: You introduce a workflow and claim that this is representative of most of software engineering - what evidence is there to reflect this? This seems very reminiscent of the issue resolution pipeline that prior works already introduce. Also, “Although different SE tasks require varied approaches, they all follow a similar workflow.” - Are there citations or prior evidence from existing SE works that reflect that the workflow you discuss is in fact the most reflective? This part of the introduction feels quite anecdotal.”
> > >
> > > We appreciate the feedback regarding the workflow presented in lines 45–104 of our manuscript. Our intention was to abstract and streamline the software engineering (SE) process into three core stages: planning, feature localization, and execution. This abstraction aligns with established SE methodologies, which often encompass similar phases.
> > > For instance, the Software Development Life Cycle (SDLC) is a well-recognized framework that includes stages such as planning, analysis, design (reasoning on current codebase), implementation (finding relevant, dependency then edit), testing (finding target function, dependencies localization then edit testing files to include new tests), deployment (can also be represented as finding relevant documents, writing deployment script, then executing deployment command e.g Azure CLI) and maintenance [6]. Our proposed workflow consolidates these stages into broader categories to emphasize the iterative and interconnected nature of SE tasks.
> > >
> > > We acknowledge that prior works have introduced similar issue resolution pipelines. Our goal was to provide a high-level abstraction that highlights the commonalities across various SE tasks, facilitating a more adaptable and scalable approach.
> > >
> > > [6] Ruparelia, Nayan B. "Software development lifecycle models." ACM SIGSOFT Software Engineering Notes 35.3 (2010): 8-13.

---

> > > > ### Author Response · Authors · 2024-11-25
> > > > **Reminder: Rebuttal Deadline Tomorrow**
> > > >
> > > > Reminder: Rebuttal Deadline Tomorrow
> > > >
> > > > Dear Reviewer,
> > > >
> > > > We hope this message finds you well. This is a gentle reminder that the rebuttal deadline is tomorrow, November 26. We wanted to check if our response has sufficiently addressed your concerns. Please don’t hesitate to reach out if there are any remaining questions or points you’d like us to clarify or discuss further.
> > > >
> > > > Best regards,
> > > > The Authors

---

> > > > > ### Author Response · Authors · 2024-12-02
> > > > > **Reminder: Rebuttal Deadline Today**
> > > > >
> > > > > Dear Reviewer,
> > > > >
> > > > > We hope this message finds you well. This is a gentle reminder that the rebuttal deadline is today, December 2. We wanted to check if our response has sufficiently addressed your concerns. Please don’t hesitate to reach out if there are any remaining questions or points you’d like us to clarify or discuss further.
> > > > >
> > > > > Best regards, The Authors

---

> > > > > > ### Comment · Reviewer_Wb3u · 2024-12-02
> > > > > > **Response to Authors**
> > > > > >
> > > > > > I greatly appreciate the author's thorough responses to my questions. I have decided to maintain my score for this work. My main remaining qualm is that while HyperAgent's performance is admirable compared to existing agentic approaches, I am of the position that these changes do not seem that different from existing prior solutions on SWE-bench. The main sell of HyperAgent being a generalizable agent, while admirable, also feels like a bit of an unfounded claim given that the evaluations are limited to SWE-bench and Defects4J. Codebase-level evaluations in different languages, although few, do exist (Multipl-E, SWE-bench Multimodal - although the second work is unpublished so I do not expect metrics for SWE-bench Multimodal), and the generalizability claim may be stronger if these evaluations are incorporated, or if the authors introduce their own.

---

> > > > > > > ### Author Response · Authors · 2024-12-03
> > > > > > >
> > > > > > > In addition to SWE-Bench and Defects4J, we would like to highlight that our evaluation also includes RepoExec for repository-level code generation. While the Multiple-E benchmark you mentioned supports multiple languages, it is not a repository/codebase-level benchmark. Our ultimate goal is to evaluate real-world systems, which necessitates using benchmarks that operate at the repository/codebase level. Please consider this aspect.

---

### Official Review · Reviewer_coAh · 2024-11-05

**Soundness:** 3
**Presentation:** 2
**Contribution:** 2
**Rating:** 5
**Confidence:** 5

**Summary:**

The paper proposes an LLM agent approach for software engineering tasks. It identifies that SE tasks often involve a combination of localisation, editing and execution, all orchestrated by some higher-level plan. A framework made up of modular components for each of these is proposed where each component is a ReAct style LLM agent loop with access to specialised tools. Through positive results on software generation (RepoExec) and bug-fixing (SWE-bench and Defects4J), the paper aims to show generalisability and efficiency (in terms of API costs).

**Strengths:**

The paper provides a clear framework for decomposing software engineering tasks and solving these in a specialised manner. Specifically, the tools proposed by the paper, as well as how they are called (writing python code) are useful contributions. The ablations show tool design adds significant value to the framework, particularly the choice of what controls are given to the LLM (e.g. code_search, keyword summaries, and repair editor). The paper provides broad evaluation on both generation tasks (RepoExec) as well as bug-fixing (Defects4J and SWE-bench) which support claims of generalisability.

**Weaknesses:**

1. The paper lacks thorough comparison to prior approaches that employ similar multi-stage approaches. CodeR [1] uses a multiagent approach for with seperate manager, localiser, editor and verifier (very similar to the components of Hyperagent), achieving 28.3% on SWE-bench-lite. MASAI [2] uses a modular architecture with similar components (without a manager) achieving 28.3% on SWE-bench-lite. Approaches such as AutoCodeRover (v20240620) (30%) [3] and Aider (26.3%) [3] both have phases for localisation and editing. Overall, the novelty of such a multi-stage approach is questionable, especially since the performance on SWE-bench is comparable to similar approaches.

2. The paper also does not provide sufficient evidence as to why a multi-stage approach in general is more "generalisable". The performance numbers quoted for agentless (24.3%) seem out of date, with Agentless + GPT 4o (2024-05-13) achieving 27.3%. While it is understandable that comparing with all new entries on swebench is unreasonable, comparing with one of the many strong non multi-stage approaches both quantitatively and qualitatively would do much to strengthen the papers claims.

In general, while the paper demonstrates positive results of their approach on two different kinds of tasks, the novelty of the approach as well as comparisons with prior work are lacking.

[1] https://arxiv.org/pdf/2406.01304
[2] https://arxiv.org/pdf/2406.11638
[3] https://www.swebench.com/

**Questions:**

1. Is the Planner sequential or can it also dispatch parallel queries to multiple agents.

---

> ### Author Response · Authors · 2024-11-20
>
> We appreciate the thoughtful feedback and the opportunity to address the concerns raised. Below, we provide detailed responses to each point.
>
> ---
> ### Comparison with CodeR and MASAI
>
> **While CodeR also employs a multi-agent framework, there are significant differences in design and execution between CodeR and HyperAgent.**
>
> #### **1. Agent Coordination:**
> - **CodeR** relies on a **rigid task graph** to enforce agent interactions, which constrains its adaptability to evolving or non-standard tasks.
> - **MASAI** employs a fixed sub-agent sequence tailored specifically for **GitHub issue resolution**, with components like a **Ranker** explicitly designed for patch proposals.
>   - These fixed designs **limit the flexibility and applicability** of both frameworks to a broader range of software engineering tasks.
> - **Other Approaches**: Systems like **AutoCodeRover** and **Aider** employ rigid, single-phase flows that limit adaptability.
>   - For example, if the localization phase in AutoCodeRover fails to identify correct buggy locations, the entire patching process fails, as it cannot revisit the localization phase.
>
> **In contrast, HyperAgent is inspired by the natural workflows of software developers, enabling flexibility and adaptability:**
> - HyperAgent encompasses four key stages:
>   1. **Analysis and Planning**,
>   2. **Feature Localization and Knowledge Grounding**,
>   3. **Code Editing**, and
>   4. **Execution.**
> - This design allows HyperAgent to **adapt easily** to diverse software engineering tasks without requiring significant redesign.
> - **Dynamic Orchestration**:
>   - HyperAgent’s **Planner** dynamically orchestrates agent interactions in real time, tailoring strategies to specific problems while maximizing reasoning capabilities.
>   - For example, if an edit fails to resolve an issue, the Planner can:
>     - Switch back to the **localization phase** using the Navigator agent, or
>     - Continue refining the edit made by the Editor.
>   - This dynamic adaptability distinguishes HyperAgent from the rigid approaches of **CodeR**, **MASAI**, **AutoCodeRover**, **Agentless**, or **Aider**.
>
> ---
>
> #### **2. Planner as a Central Component:**
> - While **CodeR’s Manager** handles user interaction and plan selection, **HyperAgent’s Planner** is integral to problem-solving.
> - The Planner performs the following:
>   - **Generates high-level strategies** to resolve core problems (e.g., identifying root causes in GitHub issue resolution).
>   - **Determines optimal subgoals**.
>   - **Selects appropriate agents** for execution.
>   - **Interprets the outcomes** of agent actions.
>
> **Optimized Communication Flow**:
> To ensure optimal performance, we designed the communication flow between the Planner and child agents with care:
> 1. **Minimized Contextual Noise**:
>    - Child agents transfer only **essential grounding information** to the Planner, avoiding noisy or lengthy contexts that could degrade LLM reasoning capabilities (as noted in [1]).
> 2. **Format-Free Reasoning**:
>    - The Planner operates without rigid agentic formats, enabling better reasoning and problem-solving (as supported in [2]).
>
> This architecture enhances HyperAgent’s adaptability, efficiency, and generalizability across tasks beyond GitHub issue resolution.
>
> ---
>
> #### **3. Fault Localization**:
> - **CodeR** relies on **BM25** and **test coverage** for fault localization, which:
>   - Requires **repository-specific configurations**.
>   - Reduces generalizability.
>   - Demands additional engineering efforts to adapt to new programming languages or scenarios.
>
> **HyperAgent abstracts navigation tasks** into Navigator agent with modular tools with **natural interfaces**:
> - Tools like **language servers** execute requests under the hood, making adaptation to new repositories straightforward.
> - This abstraction reduces engineering overhead and ensures compatibility with diverse programming languages and scenarios.
>
> ---
>
> We hope this detailed response clarifies how **HyperAgent’s dynamic, modular, and adaptive design** distinguishes it from prior frameworks like CodeR, MASAI, and other approaches. We are committed to enhancing the manuscript further to address these key differences and contributions.
>
> [1] Fraga, Natanael. "Challenging LLMs Beyond Information Retrieval: Reasoning Degradation with Long Context Windows." (2024).
>
> [2] Tam, Zhi Rui, et al. "Let me speak freely? a study on the impact of format restrictions on performance of large language models." arXiv preprint arXiv:2408.02442 (2024).

---

> > ### Author Response · Authors · 2024-11-20
> >
> > Additional point: we also pointed out the differences of other works rather than MASAI, CodeR (which are currently not open sourced) like MetaGPT (open source) with HyperAgent in our response to Reviewer Wb3u.

---

> > > ### Author Response · Authors · 2024-11-23
> > >
> > > ### Question:
> > > > Is the Planner sequential or can it also dispatch parallel queries to multiple agents."
> > >
> > > ---
> > >
> > > Yes, Planner can dispatch multiple, independent, and parallel queries to multiple agents at the same time. For example, Planner want to build requirements while editing some files or investigate the codebase, this mechanism significantly improves the execution time of the whole pipeline (on SWE-Bench, we can achieve 2-6x improvements).
> > >
> > > ---
> > > In case you need information on how HyperAgent avoid child agents conflict the same resources, Reviewer B9Au also ask a question about this mechanism
> > >
> > > ### Question:
> > > > "Can the authors clarify if there are specific design decisions made to ensure synchronization between agents preventing them from modifying the same files?"
> > > Synchronization is managed by the Planner, which centralizes task generation and distribution. For example:
> > >
> > > The Planner ensures that tasks such as editing or navigating are assigned to distinct files or independent parts of the codebase.
> > > While intelligent planners like Claude 3 Sonnet rarely produce conflicting tasks, we include an additional ad hoc check to deduplicate tasks that overlap (e.g., editing the same file or issuing redundant navigation requests).
> > > This design ensures robust task coordination and prevents errors.

---

> > > > ### Author Response · Authors · 2024-11-25
> > > > **Reminder: Rebuttal Deadline Tomorrow**
> > > >
> > > > Dear Reviewer,
> > > >
> > > > We hope this message finds you well. This is a gentle reminder that the rebuttal deadline is tomorrow, November 26. We wanted to check if our response has sufficiently addressed your concerns. Please don’t hesitate to reach out if there are any remaining questions or points you’d like us to clarify or discuss further.
> > > >
> > > > Best regards,
> > > > The Authors

---

> ### Author Response · Authors · 2024-11-20
>
> ### Comment:
> > "The paper also does not provide sufficient evidence as to why a multi-stage approach in general is more "generalisable"."
>
>
> **In our work, generalizability refers to HyperAgent’s ability to adapt seamlessly to a wide range of software engineering (SE) tasks and programming languages without requiring significant architectural changes.** Here’s how HyperAgent achieves this:
>
> ---
>
> #### **1. Modularity and Specialized Agents**:
> - HyperAgent comprises specialized agents (**Navigator**, **Editor**, **Executor**, and **Planner**), each tailored to perform distinct SE functions.
> - This modularity allows us to:
>   - Easily introduce new agents.
>   - Modify existing agents to accommodate additional tasks or programming languages.
> - **Dynamic Planning**:
>   - HyperAgent’s **Planner** provides maximized reasoning capability and dynamically orchestrates tasks, as detailed in the **Comparison with CodeR and MASAI** section above.
>
> ---
>
> #### **2. Programming Language-Agnostic Tool Interfaces**:
> - HyperAgent leverages **language servers** that provide standardized interfaces for various programming languages.
> - By abstracting tool interactions through these servers, HyperAgent can support multiple languages (e.g., Python, Java) with **minimal adjustments**.
>   - **Example**: Integrating a new language involves connecting to its corresponding language server, which handles tasks like:
>     - Linting.
>     - Code search.
>     - Definition lookups.
>
> ---
>
> #### **3. Ease of Adaptation to New Tasks**:
>
> we can introduce new SE tasks into HyperAgent involving modifying or creating task-specific prompt templates, rather than overhauling the entire system architecture.
>
> ---
>
> #### **4. Comprehensive Evaluation Across Diverse Tasks**:
> - HyperAgent has been rigorously evaluated across a variety of software engineering tasks, achieving **state-of-the-art results**:
>   - **RepoExec**: Leading performance for code generation.
>   - **Defects4J**: Demonstrated superior bug-fixing capabilities.
>   - **SWE-Bench**: Comparable performance to specialized baselines like AutoCodeRover and Agentless.
>
> ---
>
> **In summary**, HyperAgent’s modular design, language-agnostic interfaces, and task-specific adaptability enable it to handle varied SE tasks effectively, demonstrating its generalizability across tasks, languages, and scenarios. We admitted that we did use out dated performance reported in Agentless paper at the time we wrote this paper, we will update this entry in manuscript.

---

> ### Comment · Reviewer_coAh · 2024-11-26
>
> Thank-you to the authors for the response and clarifying how this work varies from similar prior approaches. However, given that this work's performance on swebench is comparable or lower than both structured and unstrctured approaches, it is still unclear whether the central claim of this paper that "flexibility and adaptability" of such an approach leads to either concrete improvements in task performance or better generalizability (without comparisons on other benchmarks with other agentic approaches).

---

> ### Author Response · Authors · 2024-11-26
>
> Thank you for your thoughtful follow-up question. We appreciate the opportunity to clarify further and address the concerns raised. Below, we provide detailed reasoning and evidence to support the central claims of **flexibility**, **adaptability**, and **generalizability** in **HyperAgent**.
>
> ---
>
> ### **Performance on SWE-Bench**
>
> While HyperAgent’s performance on SWE-Bench is comparable to other structured and unstructured approaches, we would like to emphasize several key points:
>
> 1. **Task Complexity and Stochasticity**
>   SWE-Bench is an inherently challenging benchmark due to its complex and stochastic problem-solving dynamics with some unreliable test cases, compounded by the high cost of evaluation and variability in LLM performance, which can lead to unstable statistics during execution. Slight differences in task resolution rates (e.g., between Lite versions of HyperAgent and other baselines)  can often be attributed to these factors, making it challenging to draw definitive conclusions from minor performance differences.
>
> 2. **Comparable Results with Cost Efficiency**
>    Even on SWE-Bench, HyperAgent demonstrates **comparable performance** with modular Lite configurations, offering cost savings and enabling real-world deployment flexibility.
>
> ---
>
> ### **Generalizability Beyond SWE-Bench**
>
> HyperAgent’s **core advantage lies in its flexibility and adaptability**, as demonstrated on **other benchmarks** not only SWE-Bench where it outperforms specialized systems:
>
> 1. **Program Repair (Defects4J)**
>    - HyperAgent achieved **25% accuracy** on program repair tasks, outperforming the specialized **RepairAgent** system, which employs a task graph-based approach similar to CodeR, achieving only **20.5%**.
>
> 2. **Bug Localization**
>    - HyperAgent achieved **59.70% localization accuracy**, surpassing **AutoFL** (51%), an agentic system with a fixed architecture similar to **Agentless**.
>
> ### **Challenges in Benchmark Comparisons**
>
> We acknowledge the reviewer’s concern about comparisons with other agentic systems. However, we hope the following points provide sufficient context:
>
> 1. **Limited Availability of Source Code**
>    - Systems like **CodeR** and **MASAI** have not released their source code, making direct benchmarking infeasible.
>    - This limitation prevents us from directly testing HyperAgent against these systems on broader benchmarks.
>
> 2. **Cross-Language Benchmarking Challenges**
>    - SWE-Agent and Agentless are Python-focused systems and do not natively support Java benchmarks like **Defects4J**.
>    - Adapting these systems would require significant engineering efforts to address cross-language challenges (e.g., fault localization or repair in Java). In contrast, HyperAgent easily adapts to new programming languages through its language server interface, showcasing its flexibility.

---

> ### Author Response · Authors · 2024-12-02
> **Reminder: Rebuttal Deadline Today**
>
> Dear Reviewer,
>
> We hope this message finds you well. This is a gentle reminder that the rebuttal deadline is today, December 2. We wanted to check if our response has sufficiently addressed your concerns. Please don’t hesitate to reach out if there are any remaining questions or points you’d like us to clarify or discuss further.
>
> Best regards, The Authors

---

### Official Review · Reviewer_AhW1 · 2024-11-08

**Soundness:** 3
**Presentation:** 3
**Contribution:** 3
**Rating:** 6
**Confidence:** 3

**Summary:**

This paper presents LLM based multi-agent system for various SE tasks, consisting of Navigator, Editor, Executor and Planner to control there 3 agent modules.
The paper shows the versatility of HyperAgent by applying 3 tasks, i.e. issue resolution, code generation, fault localization and repair, along with comparison with SOTA models, and how it outperforms other methods. The authors also provide a detailed analysis of agent roles, tool design and behaviour,

**Strengths:**

It showed the avility to solve multipole software engineering tasks such as issue resolution, code generation, fault localizaiton and repair. The design enable users to use lightweight (cost efficient) LLM for navigation agent and strong LLM for other agents, depending on the problem.

**Weaknesses:**

Implimantation details is not clear until the appendix. For example, Footenote 1 on  page 2 says" details of each agent are writtin in Sec4". But  Sec.4 is too short to understand. This is not "detail", at least "Implimentation summary" or "overview". Please add  more details on the architecture of each agent, their interactions, or the key algorithms/prompt used, for example.

How much the system is scalable and how did it implimented is not clear. From sec 3.2 denoted it is scalable because of using message queue. But it is not clear how it is related to scalablity. I understand that MessageQueue is designed for large system integration, but not show how it is related to "numerous subtasks, handlig complix tasks efficiently", which the authors wrote on Introduction.

**Questions:**

The result and analysis shows the developed system works well, how ever it is not clear how generalizability and scalability is presented in the paper. It is recommend that authors clearly and explicitly explain how you achieved generality, efficiency, and scalability, what experiments you conducted, and how you evaluated the result section. Or/and add  section on generalizability and scalability experiments, because as for "scalability", there are 3.2 AGENT COMMUNICATION AND SCALABILITY, but ther eare no such description on other 2 features.

---

> ### Author Response · Authors · 2024-11-20
>
> We thank the reviewers for their constructive feedback and insightful questions, which have helped us identify areas to clarify and improve. Below, we address the key concerns raised:
>
> ---
>
> ### **Implementation Details in Section 4**
>
> **We acknowledge that Section 4 would benefit from additional elaboration to provide a more comprehensive architectural overview.** While it currently summarizes agent roles, tools, and configurations, it lacks sufficient detail regarding the system’s underlying design. To address this, we propose the following clarifications:
>
> ---
>
> #### **1. Detailed Architecture**:
> The implementation details of HyperAgent are outlined in **Appendix A.7**, which includes **agent-specific prompt templates**. All agents, except the Planner, follow a **ReAct-like design** [1]. In each iteration, an agent performs the following sequence:
> - **Thought Generation**:
>   The agent generates a thought string, representing its reasoning process for the current task (e.g., *“Determine which function contains the bug”*).
> - **Action Generation**:
>   The agent produces an action string, implemented as **Python code** [2]. This code serves as the unified action space, with tools (e.g., navigation or editing functions) predefined as built-in Python functions that the agent can invoke.
> - **Observation**:
>   The agent receives an observation string, capturing the execution results of the generated Python code (e.g., the output of a code search or edit operation).
>
> **Detailed examples** of agent trajectories, including interactions and tool usage, are provided in **Section A.3**.
>
> ---
>
> #### **2. Planner’s Role**:
> The Planner operates differently, serving as the **central coordinator**. In each iteration, it generates:
> - A **thought string**, dynamically outlining its strategy to resolve the current task.
> - A **request** to a child agent (e.g., Navigator, Editor, or Executor), specifying the action required.
> - A **context**, which includes:
>   - **Objective Details**: Information necessary for the child agent to complete the specific task.
>   - **Global Background**: Broader context required for successful task execution (e.g., dependencies or impacted files).
>
> **Example**:
> If the Planner instructs the Editor to modify a Python file to add a new feature, the context might include details on the impacted files or functions required for dependency resolution. Once the child agent completes its task, it sends a **summary of its execution** back to the Planner.
>
> ---
>
> #### **3. Agent Communication**:
> The communication framework, detailed in **Section 3.2 (“Agent Communication and Scalability”)**, ensures smooth interactions between the Planner and child agents:
> - It employs an **asynchronous message queue system** to handle requests and responses, enabling **dynamic task coordination** and **efficient resource allocation**.
>
> ---
>
> #### **4. Scalability and Message Queue Role**:
> The role of the message queue in enabling scalability was briefly outlined in **Section 3.2** but requires clearer linkage to task complexity. We emphasize:
> - **Parallelism**:
>   - Multiple **Navigator** or **Executor instances** can run concurrently, handling subtasks independently.
>   - This design significantly reduces bottlenecks in large codebases or multiple user request scenarios (e.g., real deployment).
> - **Resource Allocation**:
>   - The message queue dynamically distributes subtasks, allowing efficient scaling with minimal overhead.
> - **Retries and failure determination**: Each tool of child agent explicitly reports errors (e.g., syntax issues or failed edits).
> These errors are summarized in the agent trajectory and sent back to the Planner. Then Planner will decide whether we should retry this request, if yes, this will be inserted into the queue.
>
> ---
>
> ### **Demonstrated Efficiency**:
> The scalability and efficiency of these design details are reflected in the **low average task execution times** in **SWE-Bench (Section 5.1.2)**:
> - **Full-1 Configuration** (Claude 3 Sonnet): 320 seconds.
> - **Lite Version**: 106 seconds.
> - Compared to **AutoCodeRover’s 720 seconds**, HyperAgent achieves **2–6x time efficiency**.
>
> We will expand **Section 3.2** with concrete examples to better demonstrate these capabilities.
>
> [1] Yao, Shunyu, et al. "ReAct: Synergizing Reasoning and Acting in Language Models." The Eleventh International Conference on Learning Representations.
>
> [2] Wang, Xingyao, et al. "Executable Code Actions Elicit Better LLM Agents." Forty-first International Conference on Machine Learning.

---

> ### Author Response · Authors · 2024-11-20
>
> ### Question:
> > "The result and analysis shows the developed system works well, how ever it is not clear how generalizability and scalability is presented in the paper. It is recommend that authors clearly and explicitly explain how you achieved generality, efficiency, and scalability, what experiments you conducted, and how you evaluated the result section. Or/and add section on generalizability and scalability experiments, because as for "scalability"..."
>
> The reviewer rightly pointed out the need for explicit validation of these properties. Our evaluations (e.g., RepoExec and Defects4J benchmarks) implicitly address generality and efficiency
>
> ---
>
> ### **Generality**:
>
> HyperAgent’s adaptability across diverse benchmarks, such as RepoExec and Defects4J, demonstrates its generality. It seamlessly handles tasks in multiple programming languages (e.g., Python and Java) with minimal effort, requiring only modifications to the task prompt template. Unlike systems like AutoCodeRover [3], which relies on language-specific tools tailored for Python, HyperAgent abstracts functional tool designs through a natural interface between child agents and a language server. These language servers, which handle features like linting, “go to definition,” and code search, can be easily replaced or extended to support new programming languages. This middleware abstracts the complexities of interacting with LSP backends, allowing the LLM to work with multiple programming languages without needing to comprehend the unique protocols of each (Appendix A.3.1). Since most languages have open-source language server implementations, integrating additional languages is straightforward.
>
> In contrast to task-specific designs like SWE-Agent [4], which rigidly incorporates issue reproduction before bug fixing, or Agentless [5] AutoCodeRover [3], CodeR [6], which all follow a fixed sequence of execution steps (e.g., localization, modification, testing), HyperAgent dynamically achieves high-level strategies through its intelligent Planner. The Planner leverages dynamic planning with specialized role prompts and operates with reduced cognitive overload. This is achieved by limiting its interactions to child agents rather than directly engaging with tools, which often produce lengthy outputs. This design ensures flexibility and efficiency in reasoning. Design details can be summarized as following points:
> - **Reduced Contextual Noise**: Child agents share only the critical grounding information with the Planner, eliminating unnecessary or overly lengthy inputs that could impair the LLM’s reasoning abilities, as highlighted in [7].
> - **Unconstrained Reasoning**: The Planner functions without the limitations of rigid agentic formats, enhancing its reasoning and problem-solving capabilities, as supported by [8].
>
> Moreover, adapting HyperAgent to new software engineering tasks is streamlined through the modification of task templates (as described in Section 3.4). By classifying tasks into two broad categories—Patch Tasks (requiring code edits) and Prediction Tasks (not requiring edits)—HyperAgent provides a reusable framework for most SE challenges.
>
> ### **Efficiency**:
>
> HyperAgent demonstrates significant efficiency improvements over state-of-the-art (SOTA) methods. Metrics such as reduced average execution time and token costs (as shown in Tables 1 and 2) highlight its ability to achieve competitive results with lower computational overhead. These results are further supported by the system’s modular architecture, which optimally distributes tasks among specialized agents.
>
> ### **Scalability**:
>
> Experiments with varying repository sizes and evaluation instances are done in parallel manner, as shown in fault localization and code generation tasks, illustrate scalability with asynchronous message queues. We will add these specific results in a dedicated subsection for clarity.
>
> ---
>
> [3] Yuntong Zhang, Haifeng Ruan, Zhiyu Fan, and Abhik Roychoudhury. 2024. AutoCodeRover: Autonomous Program Improvement. In Proceedings of the 33rd ACM SIGSOFT International Symposium on Software Testing and Analysis (ISSTA 2024). Association for Computing Machinery, New York, NY, USA, 1592–1604. https://doi.org/10.1145/3650212.3680384
>
> [4] Yang, John, et al. "Swe-agent: Agent-computer interfaces enable automated software engineering." arXiv preprint arXiv:2405.15793 (2024).
>
> [5] Xia, Chunqiu Steven, et al. "Agentless: Demystifying LLM-based Software Engineering Agents." CoRR (2024).
>
> [6] Chen, Dong, et al. "CodeR: Issue Resolving with Multi-Agent and Task Graphs." arXiv preprint arXiv:2406.01304 (2024).
>
> [7] Fraga, Natanael. "Challenging LLMs Beyond Information Retrieval: Reasoning Degradation with Long Context Windows." (2024).
>
> [8] Tam, Zhi Rui, et al. "Let me speak freely? a study on the impact of format restrictions on performance of large language models." arXiv preprint arXiv:2408.02442 (2024).

---

> ### Author Response · Authors · 2024-11-25
> **Rebuttal Deadline Tomorrow**
>
> Dear Reviewer,
>
> We hope this message finds you well. This is a gentle reminder that the rebuttal deadline is tomorrow, November 26. We wanted to check if our response has sufficiently addressed your concerns. Please don’t hesitate to reach out if there are any remaining questions or points you’d like us to clarify or discuss further.
>
> Best regards,
> The Authors

---

> > ### Author Response · Authors · 2024-12-02
> > **Rebuttal Deadline Today**
> >
> > Dear Reviewer,
> >
> > We hope this message finds you well. This is a gentle reminder that the rebuttal deadline is today, December 2. We wanted to check if our response has sufficiently addressed your concerns. Please don’t hesitate to reach out if there are any remaining questions or points you’d like us to clarify or discuss further.
> >
> > Best regards, The Authors

---

### Author Response · Authors · 2024-11-25

We are deeply grateful to the reviewers for their detailed and constructive feedback, which has significantly enhanced the quality of our work. We are encouraged by the reviewers’ recognition of HyperAgent’s contributions, particularly its ability to generalize across diverse software engineering tasks, its modular multi-agent design, and its effective evaluations on benchmarks like SWE-Bench, RepoExec, and Defects4J.

Several reviewers highlighted the novelty of combining specialized subagents with an asynchronous message queue system to enable parallel execution and improve task efficiency. They also appreciated the analysis of agent behavior, tool design, and the potential for HyperAgent to extend to additional tasks and languages.

We address the key concerns and suggestions raised by the reviewers below:

1. **Implementation Details and Scalability (AhW1, B9Au):**
   We recognize the need to provide a clearer explanation of HyperAgent’s architecture and scalability mechanisms. We have expanded Section 3.2 with concrete examples and detailed the role of the message queue in handling complex tasks and ensuring parallelism.

2. **Comparison with Prior Work (coAh, hBBP, gPre):**
   While HyperAgent shares similarities with systems like MetaGPT, SWE-Agent, and CodeR, it distinguishes itself through dynamic task orchestration, modularity, and scalability. We clarified these distinctions in Sections 2 and 3, emphasizing the Planner’s adaptive reasoning and the system’s versatility across diverse software engineering tasks. We also addressed concerns regarding task-specific baselines, highlighting HyperAgent’s superior generalization and scalability across multiple benchmarks.

3. **Performance and Cost Concerns (coAh, B9Au, gPre):**
   HyperAgent balances advanced reasoning and adaptability with competitive performance. While agentless systems like SWE-Agent achieve lower costs for specific tasks, HyperAgent’s multi-agent framework supports more complex workflows like repository-level code generation, bug localization and program repair. We introduced cost-optimized configurations (e.g., HyperAgent-Lite) to address cost concerns while maintaining strong performance.

4. **Fault Localization and Unit Testing (hBBP, B9Au):**
   We clarified HyperAgent’s fault localization approach and its comparison with ChatRepair. Unlike ChatRepair, which assumes prior localization, HyperAgent autonomously handles the full debugging pipeline. For unit testing, the Executor agent identifies and creates test cases as needed, which we have elaborated in Section 4.

5. **Experimental Results and Baselines (coAh, gPre):**
   To ensure a fair comparison, we included configurations that use the same base LLMs as baselines like Agentless and AutoCodeRover. Discrepancies in previously reported baseline results have been clarified, and we updated the manuscript to reflect these corrections.

6. **Generalizability and Tool Integration (AhW1, gPre):**
   HyperAgent’s generalizability stems from its modular design and integration with language server-based tools. These features enable seamless adaptation to new tasks and languages with minimal configuration. We provided additional details in Appendix A.3 on how tools like LSPToolKit align with LLM capabilities and compared them with existing approaches.

Based on the reviewers’ suggestions, we have revised the manuscript to clarify our contributions, address raised concerns, and provide additional supporting evidence. We hope these updates strengthen the case for HyperAgent as a scalable and generalist framework for real-world software engineering tasks.

---

### Author Response · Authors · 2024-11-26
**HyperAgent’s Real-World Focus and Balanced Evaluation Across Benchmarks**

In addition to the previous general response, we want to emphasize a critical focus of our work: **HyperAgent is designed to tackle realistic software engineering settings**, where tasks are defined and resolved at the **repository-level** rather than isolated, function-level examples. Realistic settings in software engineering involve challenges such as navigating large codebases, understanding interdependencies across files, and performing comprehensive testing—all of which mirror the daily workflows of professional developers. Benchmarks used to evaluate systems in such settings must reflect these complexities, going beyond competitive-level, single-file tasks like those in **HumanEval**, **HumanEvalFix**, or **BioCoder**.

### Defining Realistic Benchmarks: Repository-Level Challenges
Realistic benchmarks are those that evaluate a system’s ability to solve end-to-end tasks within a **repository context**, requiring navigation, dependency resolution, and multi-file understanding. The benchmarks we selected—**RepoExec**, **Defects4J**, and **SWE-Bench**—reflect this real-world complexity:

1. **RepoExec**
   RepoExec is one of the newest benchmarks designed for **repository-level code completion**. It evaluates tasks that require systems to generate code that integrates seamlessly into an existing codebase, navigating dependencies across files, and ensuring functionality in the broader repository context. RepoExec introduces a rigorous evaluation mechanism to verify the correctness of generated code, a critical aspect of real-world software engineering.

2. **Defects4J**
   Defects4J is a long-standing benchmark that challenges systems to localize and repair bugs in real-world Java repositories. Tasks in Defects4J require systems to work with real-world bugs in large, interconnected codebases, making it a prime example of a repository-level benchmark that prioritizes realistic settings over isolated snippets.

3. **SWE-Bench**
   SWE-Bench evaluates GitHub issue resolution, focusing on understanding and resolving practical software engineering issues. While valuable, SWE-Bench primarily addresses issue resolution within a limited scope and does not fully capture the repository-level challenges found in RepoExec and Defects4J.

### Encouraging Balanced Evaluation Across Benchmarks
We recognize the popularity and relevance of SWE-Bench, and we appreciate the reviewers’ focus on this benchmark. However, we would like to highlight that **RepoExec and Defects4J are equally complex and realistic benchmarks** that evaluate critical aspects of software engineering beyond issue resolution. RepoExec, for example, pushes systems to tackle repository-wide code completion, while Defects4J remains one of the most challenging and widely used benchmarks for fault localization and repair in real-world Java projects.

Given the diverse strengths and contributions of these benchmarks, we kindly suggest that HyperAgent’s achievements across all three benchmarks be considered with equal weight. This would provide a more balanced evaluation of its generalization and effectiveness across a broad range of realistic software engineering tasks.

### Realism in Evaluation and Multi-Language Generalization
By selecting these benchmarks, we ensure that HyperAgent is evaluated in scenarios that require handling **realistic software engineering tasks** involving **entire repositories** rather than simplistic, single-file examples. Moreover, these benchmarks span multiple programming languages—**Python** (SWE-Bench, RepoExec) and **Java** (Defects4J)—demonstrating HyperAgent’s ability to generalize across languages.

### Framework for Future Adaptation
To further promote realism and scalability, we provide an **open-source framework** that allows researchers to adapt HyperAgent to new programming languages and tasks. This contribution ensures that HyperAgent not only excels in its current benchmarks but also serves as a foundation for expanding to additional real-world challenges in software engineering.

### Evidence-Based Generalization
We are the **first to provide comprehensive, result-driven evidence** demonstrating HyperAgent’s generalization capabilities across repository-level benchmarks. While **Reviewer Wb3u** mentioned that other works might be as generalizable as HyperAgent, such claims have not been substantiated with rigorous evaluations across diverse tasks. HyperAgent validates its adaptability with evidence-based performance on **repository-level benchmarks** like RepoExec and Defects4J, setting a new standard for systems tackling real-world software engineering problems.

---

### Meta-Review · Area_Chair_AuDe · 2024-12-18

**Metareview:**

The primary finding of the paper is that an LLM agent architecture which improves benchmark performance on repository-level software engineering tasks. The primary strength is that it it is better than baselines. The primary weakness is that it is conceptually similar to prior agents, and not strong enough empirically relative to baselines to warrant publication at this stage. I worry, like other reviewers, that these slim SOTA claims are fragile, and derive more from benchmark specifics than broad scientific principles or novel conceptual insight.

**Additional Comments On Reviewer Discussion:**

Most authors engaged with the rebuttal, which was not successful at persuading either them (or AC) of the conceptual novelty or the robustness of the empirical improvements.

---

### Decision · Program_Chairs · 2025-01-22

Reject